# Considerations and Challenges in the Management of the Older Patients with Gastric Cancer

**DOI:** 10.3390/cancers14061587

**Published:** 2022-03-21

**Authors:** Sotiris Loizides, Demetris Papamichael

**Affiliations:** Department of Medical Oncology, Bank of Cyprus Oncology Centre, Nicosia 2006, Cyprus; demetris.papamichael@bococ.org.cy

**Keywords:** gastric cancer, elderly, early-stage disease, adjuvant, metastatic, targeted therapies

## Abstract

**Simple Summary:**

Older patients diagnosed with gastric cancer represent a significant proportion of patients with the disease. These patients can range from extremely fit to very frail and can have different expectations and goals from their younger counterparts. Currently there is little evidence available in the literature to guide management. The design of clinical trials specific to this age group is therefore urgently needed. In the meantime, these patients should undergo some form of geriatric screening when managed during routine clinical practice.

**Abstract:**

Gastric cancer is one of the commonest malignancies with high rates of mortality worldwide. Older patients represent a substantial proportion of cases with this diagnosis. However, there are very few ‘elderly-specific’ trials in this setting. In addition, the inclusion rate of such patients in randomised clinical trials is poor, presumably due to concerns about increased toxicity, co-existing comorbidities and impaired performance status. Therapeutic strategies for this patient group are therefore mostly based on retrospective subgroup analysis of randomised clinical trials. Review of currently available evidence suggests that older gastric cancer patients who are fit for trial inclusion may benefit from surgical intervention and peri-operative systemic chemotherapy strategies. For patients with metastatic disease, management has been revolutionized by the use of anti-HER2 directed therapies as well as immune checkpoint inhibitors with or without chemotherapy. Early data suggest that fit older patients may also benefit from these therapeutic interventions. However, once again there may be limitations in extrapolating these data to everyday clinical practice with older patients being less likely to have a good performance status and an intact immune system. Therefore, determining the functional age and not just the chronological age of a patient prior to initiating therapy becomes very important. The functional decline including reduced organ function that may occur in older patients makes the integration of some form of geriatric assessment in routine clinical practice very relevant.

## 1. Introduction

Gastric cancer is one of the commonest cancers worldwide. Based on the GLOBOCAN 2020 data, stomach cancer is the sixth most frequent neoplasm and the third most deadly cancer, with an estimated 768,793 deaths [1]. Gastric cancer incidence and mortality are highly variable by region and are dependent on diet and the prevalence of Helicobacter pylori infection. Despite an upward trend in new cases among the young, it is still considered a disease of older individuals [2]. In the UK between 2016 and 2018, approximately 50% of new cases of gastric cancer were diagnosed in people over the age of 75, while the highest rates were in the 85 to 89 age group [3]. In the US, the average age at diagnosis for stomach cancer is 68, with 6 out of 10 people diagnosed being 65 or older [4]. Increasing incidence with age presumably reflects cell DNA damage accumulating over time, resulting from biological processes and known risk factors. In addition, owing to an upsurge of life expectancy, a corresponding increase in gastric cancer cases in elderly individuals is noticed. 

Recent advances in the perioperative management of the disease, as well as better understanding of the molecular basis of gastric cancer have led to improved outcomes. However, hesitation to include older patients in clinical trials, possibly due to perceived poor treatment tolerance and lack of elderly specific trials, have left this patient population without a clear evidence basis to aid everyday management. Evidence in this setting is mostly derived from subgroup analysis of larger randomised trials and smaller prospective phase II studies.

Additional challenges for treating gastric cancer in this older age group include physiological heterogeneity, reduced treatment tolerance and different treatment goals set by patients. The aim of this article is to review the currently available evidence for older patients in the perioperative, adjuvant and palliative setting. In many studies and reviews, the age of 65 or 70 is variably used as a cut-off for describing older patients and comparing outcomes. There is currently no accepted definition for what describes an elderly patient [5]. Nevertheless, age cut-off should only exist to promote awareness, but not determine management.

## 2. Genetic and Molecular Characteristics

Over the past few years, great advances have been made towards the molecular characterisation of gastric cancer. Based on the recently described Cancer Genome Atlas (TCGA), the molecular and genetic alterations of gastric cancer can be classified into four distinct types: Epstein Barr Virus—infected tumours, tumours with Microsatellite Instability (MSI-H), tumours with Chromosomal Instability (CIN) and Genomically Stable (GS) tumours [6]. Microsatellite status is of special interest in older patients with gastric cancer. Polom et al. reported high incidence of MSI-H tumours among octogenarians [7]. In a retrospective study of 1749 patients who underwent potentially curative surgery, 433 tissue samples were retrieved and investigated for genetic alternations. In this study, older patients were more likely to have an MSI-H phenotype compared to younger patients. Furthermore, the same investigator pointed out a statistically significant correlation between microsatellite instability and older age in a meta-analysis of 48 studies [8]. In another cohort study by Mathiak et al., a subgroup analysis confirmed a statistically significant association between MSI-H status and older age [9] (Table 1). It is postulated that resected MSI-H gastric tumours may have a better outlook, possibly due to a high burden of tumour-infiltrating lymphocytes and lower rate of lymph node metastasis. In the adjuvant/neo-adjuvant setting, MSI-H phenotype may indicate a favourable prognosis, providing the potential option for omitting peri-operative or adjuvant chemotherapy. In the Medical Research Council Adjuvant Gastric Infusional Chemotherapy (MAGIC) trial, patients with MSI-H tumours who received perioperative chemotherapy had worse median Overall Survival (OS) (OS: 9.6 months, 95% CI, 0.1–22.5 months) compared to patients with MSS tumours (OS: 19.5 months, 95% CI, 15.4–35.2 months; Hazard Ratio(HR): 2.18; 95% CI, 1.08–4.42; *p* = 0.03) [10]. On the other hand, in the surgery alone arm, patients with MSI-H phenotype had a much better median OS as compared to the microsatellite-stable patients [11]. In the metastatic setting, very promising results have been reported in patients with MSI-H gastric tumours treated with immunotherapy with ages ranging from 43 to 92 years. [12]. 

### Geriatric Assessment-Identification of Frailty

Some of the key components of successful oncological treatment are the establishment of the patient’s performance status and the stratification of risks prior to therapy. However, for older patients, performance status is often not enough to assess fitness for therapy [14]. For these patients, it is vital to integrate tools in everyday clinical practice for the prediction of frailty and the identification of vulnerable individuals prior to considering treatment options [15]. Chronological age alone cannot distinguish frailty, and functional age estimated by using predictive models can be a better prognostic marker for risk stratification. A validated score to assess functional status is the Comprehensive Geriatric Assessment (CGA) [16]. CGA encompasses many aspects, such as physical function, nutrition, mental or psychological status, functional status, social support and geriatric syndromes. A meta-analysis of six studies sought to assess the role of CGA in predicting postoperative complications for older patients with gastrointestinal cancers [17]. The study enrolled 1037 patients and demonstrated that three components of CGA, comorbidity (CCI ≥ 3), polypharmacy and ADL dependency, were predictive factors for postoperative complications. Pain score above zero, polypharmacy and performance status > 0 were correlated with major morbidity after gastric surgery in a retrospective survey of 279 patients who underwent gastrectomy from 2005 to 2014 and had geriatric assessment within 30 days of surgery [18]. CGA is an effective tool for use in older patients receiving chemotherapy, especially in terms of predicting chemotherapy-induced toxicity [19]. Despite the valuable role of GCA, it can be time-consuming and a challenge to resources. Two screening tools recommended by the International Society of Geriatric Oncology (SIOG) are the G-8 and VES-13 [20]. Both are convenient and easy to use, can be reliable in clinical practice and can be completed in a short period of time. Another important factor that should be taken into consideration in these patients is sarcopenia. The prevalence of sarcopenia among patients with gastric cancer has been reported to be as high as 38%. [21]. Sarcopenia is an independent prognostic factor for severe complications after gastrectomy [22]. 

## 3. Localised Gastric Cancer

### Surgical Management

Considerable improvements in the surgical management of gastric cancer have been achieved over the last two decades. Improvements in endoscopic techniques allowing for early detection of cancer, better anaesthesiologic and minimal surgical procedures all create a better scope for the surgical management of older patients with gastric cancer. Current recommendations by ESMO [23] and Japanese guidelines [24] provide some of the best approaches for the current ‘state of the art’ management in this setting. 

Endoscopic Submucosal Dissection (ESD) is the treatment of choice for early gastric cancers that are well-differentiated, non-ulcerated, less than 2 centimetres and clearly confined to the mucosa (T1a) [23]. The incidence of nodal metastasis is negligible. ESD can be a safe treatment option in frail older patients, even in the presence of severe comorbidities. According to the Japanese guidelines, older patients with severe comorbidities and high operative risk for gastrectomy who do not completely fulfil the indications for ESD, can still be considered for the procedure [24]. Kakushima et al. demonstrated that older patients with co-existing medical problems who had an R0 resection did not have a statistically significant higher complication rate compared to younger patients [25]. With regards to adverse events, the most commonly reported complications were perforation, post-operative bleeding and rarely postoperative pneumonia [26]. Furthermore, in a study of 1188 patients with gastric cancer who underwent ESD, 459 patients were above 75 years old. The incidence rate of those adverse effects was similar to younger patients, with the exception of pneumonia, which was more common in older patients [27].

For cancers staged IB-III, surgery is necessary for potentially achieving cure. However, many surgeons are reluctant to proceed with gastrectomy in older patients due to a high burden of comorbidities, elevated perioperative risk and post-surgical complications. In the past, only a very low percentage of patients over 60 years underwent any form of surgical intervention. In England, data collected over a 25-year period (1957–1981) revealed that only 13% of patients over 80 years received any systemic treatment and less than 20% had undergone surgery [28]. However, over the last 20 years reported resection rates have increased substantially for this patient group [29]. Despite innovations in surgical techniques, the prognosis of older patients remains poorer compared to younger ones. Many surgeons are reluctant to proceed with D2 lymphadenectomy in octogenarians in view of a high prevalence of perioperative complications. In an MRC study, multivariate analysis for 5-year survival rate indicated that individuals older than 60 years old had a worse prognosis compared to those under 60 (Hazard Ratio (HR) 1.03, 95% CI 1.01–1.04, *p* = 0.0001), irrespective of the type of resection (D1 or D2 lymph-node dissection) [30]. Another study from Poland suggested a worse prognosis for older patients (>66 years), between the four groups under study as there was a statistically significant difference favouring younger patients, even if gastrectomy was successful without perioperative complications [31]. In the same study it was also shown that the rate of post-operative complications tends to be higher in older patients who undergo gastrectomy. In a Japanese study where patients were sub-divided into those over and under 70 years, there was a statistically significantly higher incidence of anastomotic leak (4.6% vs. 1.5%, *p* = 0.039) and cardiovascular complications (2.5% vs. 0%, *p* = 0.01) in the older age group [32]. The overall morbidity was higher among older patients, and an intraoperative blood loss of ≥320 mL was a significant predictive factor. Another common complication is post-operative pneumonia. Frequency of pneumonia after gastrectomy in patients aged ≥ 75 years has been reported as 5.1–13.3%, significantly higher than that of younger patients. Suzuki et al. have demonstrated that pneumonia is correlated with high postoperative mortality, prolonged hospitalisation and significantly shorter overall survival [33]. 

## 4. Perioperative (Neoadjuvant) Treatments 

Recurrent metastatic disease is still the main cause of death from GC. Hence, increasing the R0 resection rate and reducing recurrence and metastasis are some of the main goals of gastric cancer management. Perioperative chemotherapy is recommended for gastric cancers staged IB or higher [23]. In most European countries, the preferred regimen for use in the peri-operative setting in fit patients is a combination regimen of 5-Fluororuracil, Leucovorin, Oxaliplatin, Taxotere (FLOT) [34]. Benefit for the use of neoadjuvant chemotherapy was demonstrated in several clinical trials [10,34,35]; however, older individuals are mostly under-represented in such studies. The MAGIC trial divided patients into three different groups (<60 years, 60–69 years, >70 years). Adverse events by chemotherapy were comparable with other studies, and there was no statistically significant difference by age [10]. A smaller randomised phase II study explored the tolerability and feasibility of perioperative chemotherapy in potentially operable esophagogastric cancer patients over the age of 65 using FLOT or the same regimen without Taxotere ‘FLO’ [36]. The majority had an ECOG performance status of 0 or 1 (91% and 96% in FLO and FLOT, respectively). Greater toxicity was reported for the FLOT group (FLOT, 85.7% FLO, 27.3% *p* = 0.0002), but an improved median Progression-Free-Survival (PFS) (21.1 vs. 12.0 months; *p* = 0.09). Overall, twenty-nine patients out of forty-four achieved an R0 resection, 15 patients in the FLO group and 14 patients in the FLOT group. In the Japanese COMPASS trial, patients with a median age of 65.5 years were randomised to two or four courses of either S-1/cisplatin (SC) or paclitaxel/cisplatin (PC) in a two-by-two factorial design study [37]. Grade 3/4 nonhematological toxicity occurred in less than 10%, and the rate of completion of chemotherapy was 76% (31/41) in the SC arm compared to 90% (38/42) in the PC arm. In the phase III FLOT4 trial, 326 (91%) patients in the ECF/ECX group and 320 (90%) patients in the FLOT group completed all cycles of allocated preoperative chemotherapy. Age related side effects were not reported; interestingly, a trend for OS for the subgroup in patients over 70 years who received FLOT was observed, but this did not reach statistical significance (*p*-value: 0.9402 HR: 0.723) [38]. Similarly, in the CRITICS trial there was no statistically significant difference favouring chemotherapy over chemoradiotherapy in patients older than 70 years (HR: 0.81 (0.48–1.35), *p*-value: 0.065) [39] (Table 2). 

### Postoperative (Adjuvant) Treatment 

Postoperative adjuvant therapy may be considered for patients with ≥Stage IB gastric cancer and prior surgery who did not receive pre-operative chemotherapy [23]. In Asian countries, adjuvant treatment is more likely to be given after surgery and D2 lymph node dissection in patients with stage II/III gastric cancer [24]. In contrast, in North America chemoradiotherapy is a popular adjuvant treatment option. In the Adjuvant Capecitabine plus Oxaliplatin for Gastric Cancer after D2 gastrectomy (CLASSIC) trial, the use of adjuvant chemotherapy (capecitabine plus oxaliplatin) was compared to observation for patients with stage II or III gastric cancer [40]. In this study, out of 1035 patients enrolled, 269 were over the age of Disease-free survival for this subgroup favoured the chemotherapy arm (HR: 0.51 CI (0.34–0.78)), although there was no overall survival benefit (HR:0,70, (CI: 0.44–1.12)). There was no statistical analysis for treatment toxicity in relation to the older subgroup. The Japanese Adjuvant Chemotherapy Trial of S-1 for Gastric Cancer (ACTS-GS) trial, like the previous trial, did not demonstrate any OS benefit for the subgroup of patients over 70 years who received S-1 chemotherapy [41]. the authors reported that grade 3 or 4 adverse events occurred in less than 5% of the patients in the S-1 group. In a meta-analysis by Chang et al. which included CLASSIC and ACTS-GS, adjuvant chemotherapy was confirmed as having a significant impact on relapse-free survival (RFS) for older patients (HR, 0.613; 95% CI, 0.466 to 0.806), although only a marginal benefit on OS was demonstrated for the same group (HR, 0.745; 95% CI, 0.552–1.006; *p* = 0.055) [42] (Table 3). According to the authors, the somewhat disappointing results related to a small sample size representative of older individuals. In a retrospective trial which included only individuals over 70 years, 55 patients with gastric cancer stage II/III received postoperative chemotherapy after gastrectomy with D2 lymph node dissection. Statistical analysis illustrated better RFS (35.5 months, *p* = 0.042), although OS did not reach a statistically significant outcome (*p* = 0.242). Grade 3/4 adverse effects occurred in 18.2% of patients receiving chemotherapy [43]. Overall, as most of these studies were conducted in Asian populations, it may be difficult to extrapolate their outcomes in ‘Western’ older patients.

## 5. Metastatic Gastric Cancer

### 5.1. Systemic Therapy

A substantial proportion of patients diagnosed with gastric cancer present with metastatic disease. Japanese and European guidelines [23,24] suggest that fit patients with good performance status and a low burden of comorbidities can benefit from systemic chemotherapy. Treatment typically comprises doublets or triplets containing platinum and a fluoropyrimidine with or without a taxane. The FLOT65+ study which included 143 patients with measurable locally advanced or metastatic adenocarcinoma with a median age of 70 years showed lack of benefit for the triplet (FLOT) over the doublet (FLO) combination [34]. No differences were detected for patients over 70 years in relation to Progression Free Survival (PFS) and OS. The triple combination was associated with more treatment-related grade 3/4 adverse events (FLOT, 81.9%; FLO, 38.6%; *p* < 0.001) and significant deterioration of quality of life. Another study tried to address the issue of toxicity. The phase II trial ‘miniDOX’ [44] included 43 previously untreated “suboptimal” patients. It enrolled patients with ECOG PS = 2, weight loss of 10–25% and/or age ≥ 70 years. Patients included in the study received a reduced dose triplet regimen of docetaxel, oxaliplatin and capecitabine. Toxicity for the triplet regimen was high with up to 76% of patients having grade 3 to 5 adverse events. Median PFS and OS were 5.5 months and 13.5 months, respectively, comparable to a similar in design study, the phase II GATE trial [45].

In the SPIRITS phase III trial, monotherapy with S-1 was compared to S-1 plus Cisplatin, a first-line regimen commonly used in Asian countries. The median age was 62 years, and benefit in terms of OS was noted for the combination of S-1 plus Cisplatin for patients younger than 60 years. A subgroup analysis did not report any relation to toxicity with age, but more grade 3/4 adverse events including leukopenia, neutropenia, anaemia, nausea and anorexia, were observed for the combination chemotherapy group [46]. A novel, non-inferiority randomised trial, the ‘GO2’ trial funded by the Cancer Research UK, compared three dose levels of a doublet regimen (oxaliplatin, capecitabine) and explored different dose intensity of chemotherapy in terms of PFS, OS and toxicity. The median age of the participants was 77.3 years, and the statistical analysis showed no differences between the three dose levels for PFS. Level C—being the lowest—was not inferior compared to Level A—being the highest (HR = 1.10 (95% CI, 0.90–1.33)). Furthermore, patients in level C had the lowest need for dose reduction or stopping therapy due to toxicity. At the same time, they had the highest percentage of patients completing treatment without delay [47]. Importantly, the study explored geriatric assessment as a tool for treatment decision-making and used an overall treatment utility questionnaire comparing patient experience. Overall, the ‘GO2’ study suggested that reduced-intensity chemotherapy may provide satisfactory cancer control without compromising quality of life for older/frail patients with metastatic gastric cancer.

### 5.2. Targeted Agents and Immunotherapy

The monoclonal antibody trastuzumab is used in advanced or metastatic gastric cancer in combination with conventional chemotherapy in the first line setting. The ToGa trial has demonstrated a survival benefit for the use of trastuzumab when used in combination with chemotherapy [48]. Median OS was 13.8 months for the experimental arm (95% CI 12–16 months; HR 0.74; 95% CI 0.60–0.91; *p* = 0.0046), compared to 11.1 months for the chemotherapy only arm. Interestingly, a subgroup analysis suggested an advantage for those patients ≥ 60 years receiving trastuzumab. Similar survival benefit was achieved for older patients in the phase II single arm JACCRO-GC 06 trial with a median OS of 15.8 months. The regimen under investigation was a combination of trastuzumab with S-1, but without the addition of a platinum resulting in lower toxicity [49].

Ramucirumab, a fully humanised monoclonal antibody against VEGFR2, can be used in patients who have failed first line therapy for metastatic gastric cancer [50]. When used in older patients, ramucirumab appears to have a tolerable safety profile. In a subgroup analysis of the REGARD trial, patients over 65 years seemed to benefit in terms of PFS and OS as compared to placebo to a similar extent as those under Immune Checkpoint Inhibitors (ICI) have shown promising efficacy in the setting of advanced or metastatic gastric cancer; however, there is very little available evidence for the use of ICI in frail patients with a poor performance status (≥2). Theoretically, ‘immunosenescence’, the decline in the immune system occurring with older age, can reduce the efficacy of immunotherapy [51]. However, two meta-analyses exploring the role of ICIs according to age showed that the survival benefit of ICI for older patients was equal to that of younger ones [52,53]. In the CheckMate-649 trial, nivolumab showed a statistically significant improvement in OS and PFS when combined with chemotherapy in patients with advanced gastric/gastroesophageal junction (GEJ) adenocarcinoma and a Combined Positive Score (CPS) ≥ 5. In this study, 316 of 789 participants were aged ≥ 65 years and 207 of them had CPS ≥ 5. In a subgroup analysis, older patients had similar OS to younger patients [54]. The ATTRACTION-2 trial, a randomised double blind phase III trial, assessed the use of nivolumab in pre-treated patients with advanced gastric or gastroesophageal junction cancer. Subgroup analysis for patients over 65 years tended to show a benefit in terms of OS with nivolumab compared to placebo [55]. The KEYNOTE-590 trial investigated the addition of pembrolizumab to chemotherapy compared to chemotherapy alone as first-line treatment of advanced oesophageal cancer [56]. Patients over the age of 65 appeared to have a benefit in terms of PFS and OS. Older adults with good performance status generally seem to benefit similarly when treated with single-agent immune checkpoint inhibitor (ICI) therapy (i.e., PD-1 or PD-L1) to their younger counterparts [57]. However, while overall toxicity appears similar both across landmark trials and in single-institutional studies, increased hospital admissions because of poor functional status and multimorbidity in everyday clinical practice remain a challenge.

## 6. Conclusions

Older adults represent a substantial proportion of patients with stomach cancer. Those older patients who are fit enough to be enrolled into clinical trials appear to gain similar benefit from treatment to their younger counterparts. However, physiological heterogeneity, quite often reduced treatment tolerance and different treatment goals make management of such patients in everyday clinical practice very challenging. Some form of baseline geriatric health assessment in the clinic can help predict the likelihood of a good therapeutic effect without unwanted toxicity, and in this way contribute to patients’ and clinicians’ treatment decisions. Unfortunately, patients who are perhaps more frail are underrepresented in landmark clinical trials, often not reflecting ‘real world’ circumstances and the true age distribution of the disease. Tailored research in the form of carefully designed ‘elderly-specific’ trials is needed to address this evidence gap.

## Figures and Tables

**Table 1 cancers-14-01587-t001:** Studies investigating MSI status in elderly patients with gastric cancer.

Study/Year/Journal	Sample of Elderly Individuals	Outcome
Chew-Wun-Wu et al. [13], 2020 Aging	N = 248 ≥ 65 years MSI-H = 31	Positive correlation between MSI-H and old age
Cohort, M. Mathiak et al. [9], 2017 Applied Immunohistochemistry and Molecular Morphology	N = 220 ≥ 68 years MSI-H = 22	Positive correlation between MSI-H and old age

MSI: Microsatellite Instability.

**Table 2 cancers-14-01587-t002:** Data from studies regarding perioperative treatment.

Study	Sample of Elderly Individuals	Outcome
Phase III, MAGIC trial [10]	N = 105, ≥70 years N = 186, 60–69 years	No statistically significant difference between elderly and younger patients
Phase II FLOT65 trial [34]	N = 43 ≥ 65 years	PFS: 21.1 months on FLOT vs. 12 months on FLO, *p* = 0.09
Phase II, COMPASS trial [37]	Arm 1, N = 21, 66 years Arm 2, N = 20, 63 years Arm 3, N = 21, 66 years Arm 4, N = 21, 67 years	PRR: ArmA:43%, ArmB:40%, ArmC:29%, ArmD:38%
Phase III, FLOT4 trial [38]	N = 172 ≥ 70 years N = 229 60–60 years	In patients over 60 years received FLOT, noticed a favoured trend for OS
Phase III CRITICS trial [39]	N = 297 ≥ 60–69 years N = 172 ≥ 70 years	No heterogeneity in the HR for treatment effect by age, HR: 1.40 (0.93–2.10). HR: 0.81 (0.48–1.35)

PFS: Progression Free Survival, HR: Hazard Ratio, PRR: Pathological Response Rate, FLOT: 5-Fluorouracil, Leucovorin, Oxaliplatin, Docetaxel, FLO: 5-Fluorouracil, Leucovorin, Oxaliplatin.

**Table 3 cancers-14-01587-t003:** Data from studies regarding postoperative treatment.

Study	Sample of Elderly Individuals	Outcome
Phase III CLASSIC trial [40]	N = 269 ≥ 65 years	No statistically significance outcome favours adjuvant therapy, HR 0.70 (0.44–1.12)
Phase III ACTS-GS trial [41]	N = 408, 60–69 years N = 257, 70–80 years	No statistically significance outcome favours adjuvant therapy for patients over 60 years
Meta-analysis, Chang et al., 2017, CRT Journal [42]	N = 930 ≥ 60 years	No statistically significant outcome, but favours adjuvant chemotherapy HR: 0.745 (0.552–1.006), *p* = 0.055

HR: Hazard Ratio.

## Data Availability

Not applicable.

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
