# Peer review of "Considerations and Challenges in the Management of the Older Patients with Gastric Cancer"

_cancers, 2022, doi:10.3390/cancers14061587_

Round 1

Reviewer 1 Report

Sotiris Loizides and Demetris Papamichael performed a review to highlight the 'Considerations and challenges in the management of the older 
patients with gastric cancer'. The review attracts the attention of clinicians to the management of older gastric cancer patients, a field that could be applied to other cancer types as well. 

My only point regarding this review is that how the authors define the 'old age'? The literature review reported by the authors covers different age groups, even at 60s, which is not really considered 'old age' for getting a cancer. We know that risk of cancer increase with age, so it is not very clear what age range is the intention of the authors, we see multiple mentions of 'older patients' without knowing older than what age. It would be useful if the authors provide some introduction to the average age onset of gastric cancer and also to mention the age groups while citing references, in which the compared age groups are defined (such as Ref 5, and Ref 12). 

Author Response

Thank you for the comment. We have now changed the manuscript by adding the following in the text:

In the UK between 2016-2018 approximately 50% of new cases of gastric cancer were diagnosed in people over the age of 75, while the highest rates were in the 85 to 89 age group (lines 47-48)

Citation:

https://www.cancerresearchuk.org/health-professional/cancer-statistics/statistics-by-cancer-type/stomach-cancer/incidence#heading-One

you will find the above link in our manuscript on reference [3]

In the US, the average age at diagnosis for stomach cancer is 68, with 6 out of 10 people diagnosed being 65 or older (lines 48-49)

https://www.cancer.org/cancer/stomach-cancer/about/key-statistics.html#:~:text=Who%20gets%20stomach%20cancer%3F,year%20are%2065%20or%20older.

You will find the above link in our manuscript on reference [4]

Increasing incidence with age presumably reflects cell DNA damage accumulating over time resulting from biological processes and known risk factors. In addition, owing to an upsurge of life expectancy a corresponding increase in gastric cancer cases in elderly individuals is noticed (lines 50-52)

In many studies and reviews the age of 65 or 70 is variably used as a cut off for describing older patients and comparing outcomes (lines 61-63, prior to reference 5, now 7)

with ages ranging from 34 to 92 years (line 89, prior to reference 12, now 13).

Reviewer 2 Report

Dear authors,

I checked some plagiarism for paper. Please corret them:

-lines 168-171 from 32. reference

-lines 195-197 from 35.reference.

-line 297 from 49.reference

-lines 305 and 310 from 53.reference

Author Response

’Dear authors,

I checked some plagiarism for paper. Please correct them.’’

Thank you for this comment. It was not our intention to copy any of the text from the referenced papers, but may have inadvertently written similar sentences in our effort to report the results of those particular studies. Please find below the changed sentences as per your recommendations.

-plagiarism from Reference 32

there was a statistically significantly higher incidence of anastomotic leak (4.6% vs. 1.5%, P=0.039) and cardiovascular complications (2.5% vs. 0%, P=0.01) in the older age group [31]. The overall morbidity was higher among older patients and an intraoperative blood loss of ?320 mL was a significant predictive factor.

-plagiarism from reference 36

Overall, twenty-nine patients out of forty-four achieved an R0 resection, 15 patients in the FLO group and 14 patients in the FLOT group.

-plagiarism from reference 50

Ramucirumab, a fully humanised monoclonal antibody against VEGFR2 can be used in patients who have failed first line therapy for metastatic gastric cancer [49]. When used in older patients, ramucirumab appears to have a tolerable safety profile. In a subgroup analysis of the REGARD trial, patients over 65 years seemed to benefit in terms of PFS and OS as compared to placebo, to a similar extent as those under 65.

-plagiarism from reference 54

In the CheckMate-649 trial, nivolumab showed a statistically significant improvement in OS and PFS when combined with chemotherapy in patients with advanced gastric / gastroesophageal junction (GEJ) adenocarcinoma and a Combined Positive Score (CPS) ≥5.

Reviewer 3 Report

This study needs the authors to critically assess the reported literature and give recommendations for elderly (>70 years) patients with gastric cancer.  From each of the trials and surgery reported, it appears that elderly patients do as well as younger patients.  The authors hypothesize that elderly patients do worse and that they need to be treated differently.  However, the data that they present do not support that argument.  I agree that the authors are correct assessing the elderly and how they would undergo chemotherapy, immunotherapy and surgery.  But if the assessment suggests that they re similar in ability to tolerate it as younger patients than they need to be treated the same (usually preoperative FLOT or FLO plus surgery).   We are not certain that older or frail patients are not included in perioperative trials so the authors cannot conclude that that is the case.  The conclusion needs to be changed based on the fact that older patients did as well as younger patients in most trials.

Author Response

‘’This study needs the authors to critically assess the reported literature and give recommendations for elderly (>70 years) patients with gastric cancer.  From each of the trials and surgery reported, it appears that elderly patients do as well as younger patients.  The authors hypothesize that elderly patients do worse and that they need to be treated differently.  However, the data that they present do not support that argument.  I agree that the authors are correct assessing the elderly and how they would undergo chemotherapy, immunotherapy and surgery.  But if the assessment suggests that they re similar in ability to tolerate it as younger patients than they need to be treated the same (usually preoperative FLOT or FLO plus surgery).   We are not certain that older or frail patients are not included in perioperative trials so the authors cannot conclude that that is the case.  The conclusion needs to be changed based on the fact that older patients did as well as younger patients in most trials.’’

Thank you for this comment. The aim of our concluding remark was to highlight the fact that while older patients fit enough to enter clinical trials appear to benefit as much as younger patients, this may not reflect ‘real world’ circumstances where patients may not be so fit. In addition, representation of older patients in such trials is poor and does not reflect the incidence of the disease. In the practice changing MAGIC and FLOT trials the median age of patients enrolled was 62 and patients over 70 represented around 20% of the total trial patients.
We therefore suggest that the conclusion is now changed by adding the following sentence in the text:

Those older patients who are fit enough to be enrolled into clinical trials appear to gain similar benefit from treatment as their younger counterparts (lines 314-316).

Unfortunately, patients who are perhaps more frail are underrepresented in landmark clinical trials, often not reflecting ‘real world’ circumstances and the true age distribution of the disease(lines 320-322).

Round 2

Reviewer 3 Report

Accept revised work